# Structural and functional characterisation of ATF2 nuclear import reveals paralogue-selective importin-α recognition and a non-canonical NLS formed in trans

Seyed Mohammad Ghafoori[1,2], Silvia Pavan[3], Trinh Xuan Duc[3] , Sepehr Nematollahzadeh[3], Gayle F Petersen[4] , Gualtiero Alvisi[3] , Jade K Forwood[2,4]

Activating transcription factor 2 (ATF2) is a member of the AP-1 superfamily that regulates essential cellular processes through its activity as a nuclear transcription factor. Although ATF2 plays well-established roles in neurodevelopment, inflammation, and cancer, the mechanisms underlying its nuclear localisation remain poorly characterised. Here, we investigate the structural and functional basis of ATF2 nuclear import via the classical importin-α/β1 (IMPα/β1) pathway. Using quantitative in vitro binding assays, we demonstrate that ATF2 interacts with multiple IMPα paralogues. Fluorescence polarisation measurements reveal the highest binding affinity for IMPα1, with progressively weaker interactions observed for IMPα3, IMPα5, and IMPα7. Crystallographic analysis of ATF2 bound to IMPα1 identifies two basic clusters that are important for interaction: site 1 ([353]EKRRK[357]), which binds the major site of IMPα1, and site 2 ([372]KRK[374]), which binds the minor site. Mutation of key residues confirms the importance of both motifs, with site 1 contributing more substantially to binding. Quantitative confocal laser scanning microscopy analysis in HEK293A cells supports these findings, showing that mutation of both clusters is required to fully abolish ATF2 nuclear localisation. Inhibition of classical nuclear import using Bimax2 significantly reduces nuclear accumulation, whereas treatment with leptomycin B confirms a role of chromosomal region maintenance 1 (CRM1)–mediated nuclear export. Notably, ATF2 mutants incapable of nuclear import can localise to the nucleus when co-expressed with c-Jun, indicating that c-Jun can facilitate ATF2 nuclear import via heterodimerisation. Together, these results establish that ATF2 enters the nucleus through IMPα recognition of two basic clusters and highlight the redundancy and complexity of ATF2 nuclear trafficking mechanisms.

## Introduction

Activating transcription factors (ATFs), also referred to as cAMP-dependent transcription factors, constitute a subgroup of 12 members (five subfamilies) within the activator protein 1 (AP-1) transcription factor superfamily (1). ATF proteins are capable of forming homodimers or heterodimers with members of the Jun, Fos, or Maf transcription factor subfamilies, generating complexes that govern a range of cellular processes, including stress responses, embryogenesis, pathogenesis, and programmed cell death (2, 3).

ATF2 is a member of the ATF subfamily and consists of 505 amino acids. It is expressed in a wide range of tissues, with particularly high levels in the brain. Structurally similar to other AP-1 family members, ATF2 possesses a basic leucine zipper (bZIP) domain located at its C terminus. The bZIP is further divided into a basic domain (BD; residues 349–374), which is rich in basic amino acids and is critical for DNA binding, nuclear localisation, and nuclear export, and a leucine zipper (residues 375–417), which mediates homo- and heterodimerisation with other AP-1 family members, thereby facilitating its diverse transcriptional regulatory functions (2).

The classical nuclear import pathway is an energy-dependent process facilitated by seven human importin-α (IMPα) paralogues and importin-β1 (IMPβ1) (4, 5, 6). Cargo proteins containing nuclear localisation signals (NLSs) interact with specific IMPαs, forming an IMPα–cargo complex (4). This complex subsequently associates with IMPβ1 to form the trimeric nuclear import complex (5). The trimer is translocated into the nucleus via the nuclear pore complex, where RanGTP mediates its disassembly, resulting in cargo release (6). Structurally, IMPαs comprise two major domains, an N-terminal IMPβ-binding domain, and a C-terminal NLS-binding domain, which consists of 10 armadillo (ARM) repeats (7, 8).

Many cargo proteins exhibit paralogue-specific binding to IMPαs. For instance, the Ran nucleotide exchange factor RCC1 and

[1]School of Life and Environmental Sciences, University of Sydney, Sydney, Australia [2]School of Agricultural, Environmental and Veterinary Sciences, Faculty of Science and Health, Charles Sturt University, Wagga Wagga, Australia [3]Department of Molecular Medicine, University of Padova, Padova, Italy [4]Gulbali Institute, Charles Sturt University, Wagga Wagga, Australia

Correspondence: jforwood@csu.edu.au

HIV-1 integrase preferentially interact with IMPα3 ([9], [10]), STAT1 and Ebola virus VP24 selectively bind to IMPα5 ([11], [12]), and both SOX2 and Hendra virus W protein show specificity towards IMPα3 ([13], [14]). In contrast, other cargoes such as MERS-CoV ORF4b exhibit high-affinity binding across all IMPα paralogues ([15]). Although nuclear translocation of ATF2 has been explored previously ([2]), the underlying structural basis and the exact sequences driving such process remain poorly characterised. This is largely due to complexities such as the presence of multiple basic motifs within ATF2 and the potential for co-translocation with other AP-1 family members, such as c-Jun. In the present study, we employ a combination of biochemical, biophysical, and cell-based methods to reveal the molecular mechanisms governing ATF2 nuclear localisation and the NLS underpinning this process.

# Results

### Paralogue-selective IMPα recognition by ATF2 basic domain

To investigate the interaction between the ATF2 bZIP domain (Fig 1A) and components of the classical nuclear import pathway, we performed a nickel bead pull-down assay. We assessed binding to IMPα paralogues across all three subfamilies, IMPα1 (subfamily α1), IMPα3 (subfamily α2), and IMPα5 and IMPα7 (both from subfamily α3), as well as IMPβ1. ATF2 bZIP interacted with all tested IMPαs, whereas its interaction with IMPβ1 was substantially weaker and barely detectable (Figs 1B and S1).

To further characterise the interactions between ATF2 and IMPαs, we performed quantitative binding assays using FP with FITC-labelled peptides corresponding to the ATF2 BD (Fig 1A). These assays revealed that ATF2 exhibits the highest binding affinity for IMPα1, with an apparent dissociation constant ($K_D$) of ~85 nM. We observed weaker interactions with other paralogues, including IMPα3 ($K_D$ ~ 870 nM), IMPα5 ($K_D$ ~ 3,182 nM), and IMPα7 ($K_D$ ~ 8,306 nM) (Fig 1C).

To elucidate the binding mechanism between ATF2 and IMPα, we determined the high-resolution crystal structure of the ATF2-IMPα complex using X-ray crystallography. We employed the mouse homologue of human IMPα1 (also known as IMPα2), which shares 94.5% sequence identity with its human counterpart. IMPα2 paralogue was selected because subfamily 1 displayed the highest binding affinity, it crystallises readily, and attempts to crystallise ATF2 BD with any of the other IMPαs were unsuccessful. The structure was solved to a resolution of 2.2 Å (Table 1).

The crystal structure revealed that the ATF2 BD bound to IMPα at two distinct sites. The first (N-terminally located) basic cluster of ATF2, comprising residues [353]EKRRK[357], occupied the major binding site of IMPα, spanning ARM repeats 2–4 (Figs 1D and S2). The second basic cluster, encompassing residues ([372]KRK[374]), engaged the minor binding site of IMPα, located across ARM repeats 6–8 (Figs 1D and S2). Although this dual-site interaction superficially resembles classical bipartite NLS recognition by IMPα, the relative orientation of the two basic clusters suggests they do not originate from a single linear sequence. In canonical bipartite NLSs, the N-terminal basic cluster typically binds the minor site of IMPα, and the C-terminal basic cluster occupies the major site. In contrast, the ATF2 basic clusters displayed the reverse orientation, with the N terminus bound at the major site and the C terminus bound at the minor site, indicating that they are unlikely to originate from a single continuous NLS. Thus, this arrangement suggests that the two basic clusters are contributed by separate ATF2 molecules, consistent with ATF2 forming homodimers via its leucine zipper dimerisation domain, with each monomer contributing one basic cluster to the interface.

The ATF2 bound at the major site follows the consensus sequence X-K-[K/R]-X-[K/R]-X. It also exhibits a canonical Lys[354] at the P2 position, forming three hydrogen bonds and a salt bridge with IMPα residues Asp[192], Gly[150], and Thr[155]. The ATF2 Glu[353] interacts with residues at the P1 site, including Asn[235] and Arg[238], through hydrogen bonding and salt bridges. At the P3 site, Arg[355] forms hydrogen bonds with Asn[188], Asp[270], and Trp[184], and also engages in a salt interaction. In addition, Arg[356] and Lys[357] interact with residues at the P4 and P5 sites, including Leu[104], Arg[106], and Glu[107], as well as Ser[105], Trp[142], Asn[146], and Gln[181], respectively. Details of these interactions are presented in Fig 1D and Table 2.

At the minor site, ATF2 residues Lys[372], Arg[373], and Lys[374] interact with the P1′, P2′, and P3′ positions, respectively (Fig 1D). This binding pattern aligns with the consensus sequence for minor site recognition, [K/R]-[K/R]-X-X. In this interaction, ATF2 Lys[372] forms hydrogen bonds with IMPα Val[321], Asn[361], and Thr[328]. ATF2 Arg[373] engages with Asn361, Gln[396], Ser[360], and Trp[357] through hydrogen bonding, and forms three salt bridges with Glu[396] at the P2′ site. In addition, Lys[374] forms three hydrogen bonds with P3′ site residues Gly[281], Asn[283], and Thr[322] (see Table 2 for details).

### Structure-guided mutagenesis reveals key determinants of ATF2 binding to IMPα paralogues

Based on structural insights from our X-ray crystallographic analysis, which revealed that ATF2 binds IMPα using two distinct sites, we designed three mutant peptides to assess the functional contribution of each site (Fig 2A). These included single mutations at site 1 and site 2, as well as a double mutant targeting both sites simultaneously. FP-binding assays between these peptides and four IMPα paralogues demonstrated that mutations at either site reduced binding affinity, whereas the simultaneous mutation at both sites abolishes binding completely (Fig 2B–E). Importantly, mutation at site 1 resulted in a more pronounced loss of affinity than mutation at site 2, indicating a dominant role of site 1 in mediating ATF2-IMPα interaction.

The most significant reductions in binding affinity were observed for IMPα1 and IMPα3. Mutation at site 1 almost completely abolished binding to IMPα1 and IMPα3. In contrast, mutation at site 2 resulted in more moderate reductions of ~5-fold and ~7-fold for IMPα1 and IMPα3, respectively, highlighting the critical role of site 1 in mediating high-affinity interactions with these paralogues.

### ATF2 nuclear import is mediated by the classical IMPα/β1-dependent pathway

Based on the hypothesis that ATF2 localises to the nucleus via the classical IMPα/β1-dependent nuclear import pathway, we set out to assess this in a cellular context. HEK293A cells were transfected with GFP-tagged full-length ATF2, either alone or in combination

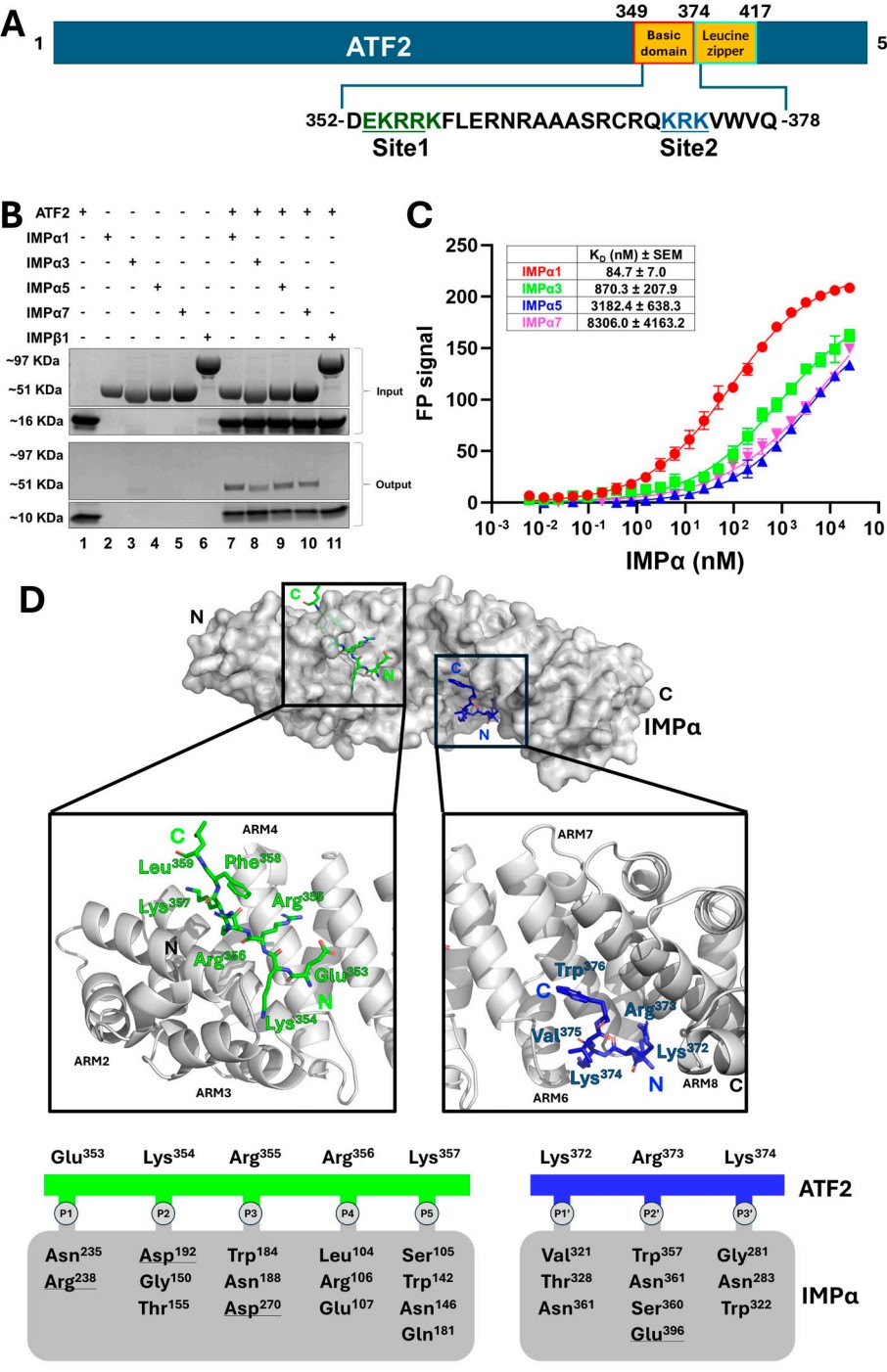

**Figure 1. ATF2 preferentially binds IMPα paralogues over IMPβ1, with specificity for IMPα1.**

**(A)** Schematic representation of the ATF2 protein, highlighting the locations of the bZIP and basic domains. The synthetic peptide sequence used for the FP assay and crystallisation is shown in one-letter code. **(B)** Nickel bead pull-down assay between the ATF2 bZIP domain and various IMPs. The assay was performed using His-tagged bZIP (~16, ~10 kD after TEV cleavage) to pull down ΔIBB IMPαs (~51 kD) and IMPβ1 (~97 kD). Lane 1 shows the ATF2 control. Lanes 2–6 are individual controls for IMPs. Lanes 7–11 show complexes formed between ATF2 and each IMP. Input (before washing) and output (after washing) samples are displayed in the upper and lower panels, respectively. **(C)** Fluorescence polarisation assays measuring the binding affinity between FITC-labelled ATF2 BD and IMPα paralogues. Dissociation constants ($K_D$ ± SEM) are shown in the corresponding table. **(D)** Structure of the ATF2 BD in complex with IMPα2. The crystal structure indicates that two molecules of ATF2 bind IMPα2 in a monopartite fashion, with site 1 of one molecule engaging the IMPα2 major site (green) and site 2 of another molecule engaging the IMPα2 minor site (blue). Key ATF2 residues include Lys354, which occupies the P2 position in the major site, and Arg373, which occupies the P2' position in the minor site. Interacting residues are shown in the schematic below, detailing hydrogen bonds and salt bridges (underlined).

with mCherry-tagged Bimax2, which has been extensively used in the field to inhibit IMPα/β1-dependent nuclear transport without affecting alternative import pathways (17, 18, 19, 20). HCMV UL44, a viral protein known to rely on the classical IMPα/β1 import pathway (21), was included as a positive control (Fig 3A). Our analysis revealed that in the absence of mCherry-Bimax2, both ATF2 and UL44 predominantly localised to the nucleus (Fig 3B), with mean nuclear-to-cytoplasmic fluorescence intensity ratios (Fn/c) of 4.4 and 25.7, respectively (Fig 3C and D). However, although UL44 accumulated in the nucleus in 100% of cells analysed, ATF2 subcellular localisation was more heterogeneous, despite accumulating in the nucleus in c. 75% of cells analysed (Fig 3C). Co-expression of mCherry-Bimax2 caused a significant reduction in nuclear accumulation of both proteins (P < 0.0001). For UL44, the

**Table 1.** Data collection and refinement statistics for the structure of IMPα2 in complex with ATF2 BD.

| IMPα2: ATF2 BD (PDB ID: 9Y0R) | |
|---|---|
| Data collection | |
| Wavelength (Å) | 0.95373 |
| Data-collection temperature (K) | 100 |
| Detector type | DECTRIS EIGER X 16M |
| Detector | PIXEL |
| Resolution range (Å) | 29.62–2.20 (2.7–2.20) |
| Space group | P 21 21 21 |
| Unit cell (Å) | 78.61 90.12 100.73 |
| Total reflections | 198,342 (16,439) |
| Unique reflections | 37,013 (3,148) |
| Multiplicity | 5.4 (5.2) |
| Completeness (%) | 99.9 (100) |
| Mean I/σ (I) | 7.6 (2.2) |
| Wilson B-factor (Å$^2$) | 31.07 |
| $R_{pim}$ | 0.071 (0.799) |
| Refinement | |
| $R_{work}$ | 0.1913 (0.2560) |
| $R_{free}$ | 0.2216 (0.2858) |
| No. of non-hydrogen atoms | 3,579 |
| No. of macromolecules | 3,353 |
| No. of solvent | 226 |
| No. of protein residues | 434 |
| Bond-length r.m.s.d (Å) | 0.002 |
| Bond-angle r.m.s.d (°) | 0.50 |
| Ramachandran favoured (%) | 98.36 |
| Ramachandran allowed (%) | 1.64 |
| Ramachandran outliers (%) | 0.00 |

Highest resolution shell statistics are shown in parenthesis.

Fn/c ratio dropped to 0.2, indicating near-complete inhibition of nuclear import. Similarly, ATF2 nuclear accumulation was significantly impaired; however, it still accumulated in the nucleus in c. 25% of cells analysed, with a Fn/c of 2.0. These data demonstrate that ATF2 nuclear import is mediated by the classical IMPα/β1 pathway, consistent with the mechanism inferred from our structural and in vitro binding studies.

## ATF2 nuclear accumulation is dynamically regulated by CRM1-dependent export

To determine whether nuclear export contributes to the steady-state localisation of ATF2, we transfected HEK293A cells with ATF2 and quantified nuclear accumulation in the presence and absence of LMB, a well-characterised inhibitor of CRM1 (exportin-1).

**Table 2.** Hydrogen bond and salt bridge interactions in the IMPα2: ATF2 BD structure.

| Hydrogen bonds | | | |
|---|---|---|---|
| ## | ATF2 | [Å] | IMPα2 |
| 1 | B:LYS 354[NZ] | 3.00 | A:ASP 192[OD1] |
| 2 | B:LYS 354[NZ] | 3.12 | A:GLY 150[O] |
| 3 | B:LYS 354[NZ] | 2.90 | A:THR 155[OG1] |
| 4 | B:ARG 355[N] | 2.77 | A:ASN 188[OD1] |
| 5 | B:ARG 355[NH2] | 3.36 | A:ASP 270[OD2] |
| 6 | B:ARG 356[NH1] | 2.90 | A:LEU 104[O] |
| 7 | B:ARG 356[NH1] | 2.86 | A:ARG 106[O] |
| 8 | B:ARG 356[NH2] | 2.99 | A:ARG 106[O] |
| 9 | B:ARG 356[NH2] | 3.65 | A:GLU 107[O] |
| 10 | B:LYS 357[N] | 2.96 | A:ASN 146[OD1] |
| 11 | B:LYS 357[NZ] | 2.96 | A:GLN 181[OE1] |
| 12 | B:LYS 372[NZ] | 2.69 | A:VAL 321[O] |
| 13 | B:LYS 372[NZ] | 2.85 | A:ASN 361[O] |
| 14 | B:LYS 372[NZ] | 2.88 | A:THR 328[OG1] |
| 15 | B:ARG 373[N] | 2.78 | A:ASN 361[OD1] |
| 16 | B:ARG 373[NH1] | 2.98 | A:GLU 396[OE1] |
| 17 | B:ARG 373[NH1] | 3.04 | A:SER 360[OG] |
| 18 | B:ARG 373[NH2] | 3.00 | A:GLU 396[OE2] |
| 19 | B:ARG 373[NH2] | 3.02 | A:GLU 396[OE1] |
| 20 | B:LYS 374[NZ] | 2.95 | A:GLY 281[O] |
| 21 | B:LYS 374[NZ] | 2.98 | A:ASN 283[OD1] |
| 22 | B:LYS 374[NZ] | 2.97 | A:THR 322[OG1] |
| 23 | B:GLU 353[O] | 3.18 | A:ASN 235[ND2] |
| 24 | B:GLU 353[OE2] | 3.32 | A:ARG 238[NH2] |
| 25 | B:GLU 353[OE2] | 3.31 | A:ARG 238[NH1] |
| 26 | B:ARG 355[O] | 2.95 | A:ASN 188[ND2] |
| 27 | B:ARG 355[O] | 2.84 | A:TRP 184[NE1] |
| 28 | B:LYS 357[O] | 3.13 | A:SER 105[OG] |
| 29 | B:LYS 357[O] | 3.21 | A:TRP 142[NE1] |
| 30 | B:LYS 357[O] | 2.84 | A:ASN 146[ND2] |
| 31 | B:ARG 373[O] | 3.01 | A:TRP 357[NE1] |
| 32 | B:ARG 373[O] | 3.02 | A:ASN 361[ND2] |
| Salt bridges | | | |
| 1 | B:LYS 354[NZ] | 3.00 | A:ASP 192[OD1] |
| 2 | B:ARG 355[NH2] | 3.36 | A:ASP 270[OD2] |
| 3 | B:ARG 373[NH1] | 2.98 | A:GLU 396[OE1] |
| 4 | B:ARG 373[NH2] | 3.00 | A:GLU 396[OE2] |
| 5 | B:ARG 373[NH2] | 3.02 | A:GLU 396[OE1] |
| 6 | B:GLU 353[OE2] | 3.32 | A:ARG 238[NH2] |
| 7 | B:GLU 353[OE2] | 3.31 | A:ARG 238[NH1] |

The interaction table was prepared using Proteins, Interfaces, Structures and Assemblies (PISA) server (16).

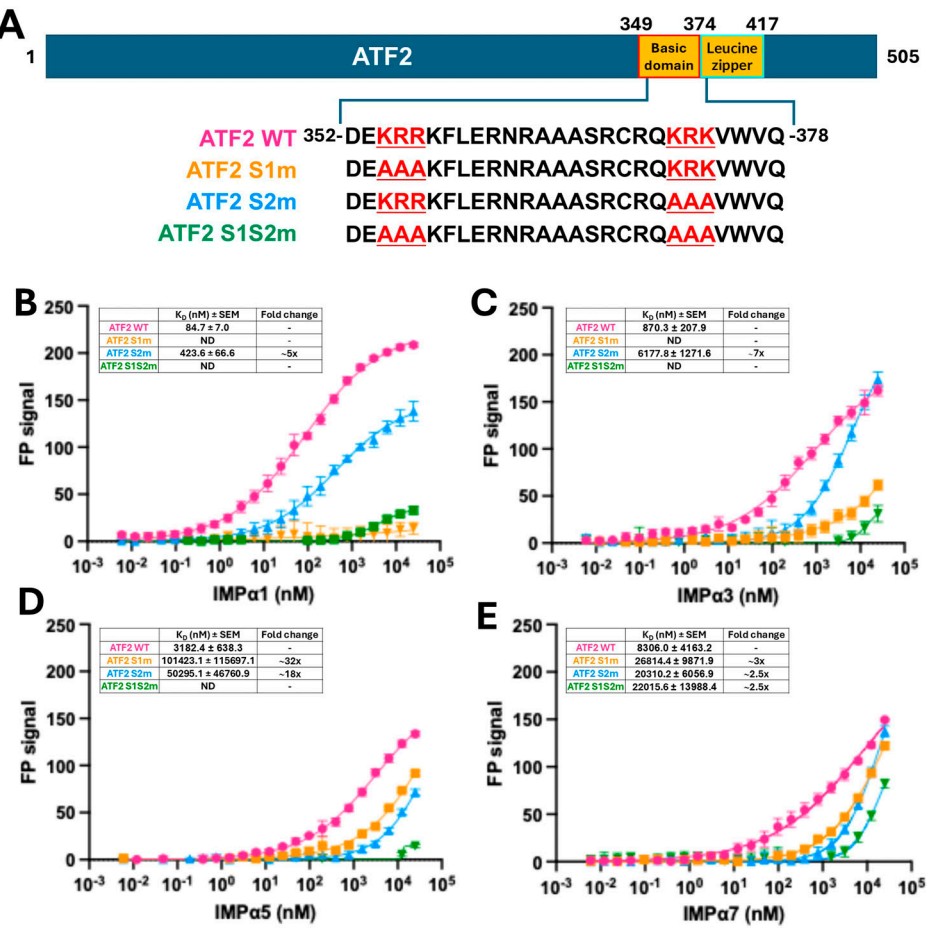

**Figure 2. Site-1 NLS plays a crucial role in mediating ATF2 interaction with IMPα paralogues.**
**(A)** Schematic of WT and mutated ATF2 BD peptides. **(B, C, D, E)** Fluorescence polarisation assays measuring the binding affinity between WT and mutant FITC-labelled ATF2 BD and IMPα paralogues. Dissociation constants ($K_D$ ± SEM) and fold change in $K_D$ for each mutant compared with the WT are shown in the corresponding table; ND, not determined. For low-affinity mutant interactions where saturation was not fully achieved, the precision of the derived $K_D$ values is low, and these values are intended primarily for comparative assessment of relative binding losses to the WT rather than as precise absolute values.

As controls, we included the HIV-1 Rev protein, which contains both a CRM1-dependent nuclear export signal (NES) and a classical NLS, and HCMV UL44 ΔNLS, a mutant lacking nuclear import capability (Fig 4A). In the absence of LMB, Rev displayed low nuclear accumulation (mean Fn/c = 0.40), consistent with active nuclear export. Upon LMB treatment, Rev localised almost exclusively to the nucleus (Fig 4B), with a marked increase in Fn/c to 25.88 (Fig 4C and D), confirming the assay's sensitivity to CRM1 inhibition. In contrast, UL44 ΔNLS showed minimal nuclear localisation both before and after LMB treatment (Fn/c = 0.3 in both cases), as expected for a protein lacking a functional NLS and thus incapable of entering the nucleus. Strikingly, LMB treatment significantly increased nuclear accumulation of ATF2, with the mean Fn/c rising from 4.4 to 7.4 (P < 0.001). These data indicate that in addition to being imported via the classical IMPα/β1-dependent pathway, ATF2 is subject to active nuclear export via the CRM1-dependent pathway, highlighting that ATF2 subcellular localisation is dynamically regulated by both nuclear import and export mechanisms.

### Site 1 is the primary nuclear import determinant of ATF2

To determine the functional contribution of each of the two basic clusters, we generated four full-length GFP-tagged ATF2 mutants:

S1m ($^{354}$KRR$^{356}$ mutated to AAA), S2m ($^{372}$KRK$^{374}$ mutated to AAA), a double mutant S1S2m (both clusters mutated), and a BD deletion construct (ΔBD; residues 348–377 deleted) (Fig 5A). Quantitative CLSM analysis confirmed that mutations at either site reduced nuclear accumulation (Figs 5B and S3). A greater reduction was observed for S1m compared with S2m: GFP-ATF2 S1m was mainly retained in the cytoplasm (Fn/c < 1) in 68% of transfected cells, compared with only 53% expressing GFP-ATF2 S2m (Fig 5B), resulting in average Fn/c of 1.2 and 1.8, respectively. Overall, these results suggest that site 1 might contribute more strongly to nuclear import. The double mutant (Fn/c = 0.5) and the ΔBD construct (Fn/c = 0.3) both showed near-complete loss of nuclear localisation (Fig 5B–D), indicating that both sites are required for efficient import.

### ATF2 can be co-transported into the nucleus with c-Jun

To investigate whether ATF2 can be translocated into the nucleus via interaction with its dimerisation partner c-Jun, we co-transfected HEK293A cells with either GFP-tagged ATF2 ΔBD or the ATF2 double mutant S1S2m, together with RFP-tagged c-Jun. The UL44 ΔNLS construct was included as a negative control. When expressed alone, all proteins showed weak nuclear accumulation, with mean Fn/c values between 0.2 and 0.5, consistent with

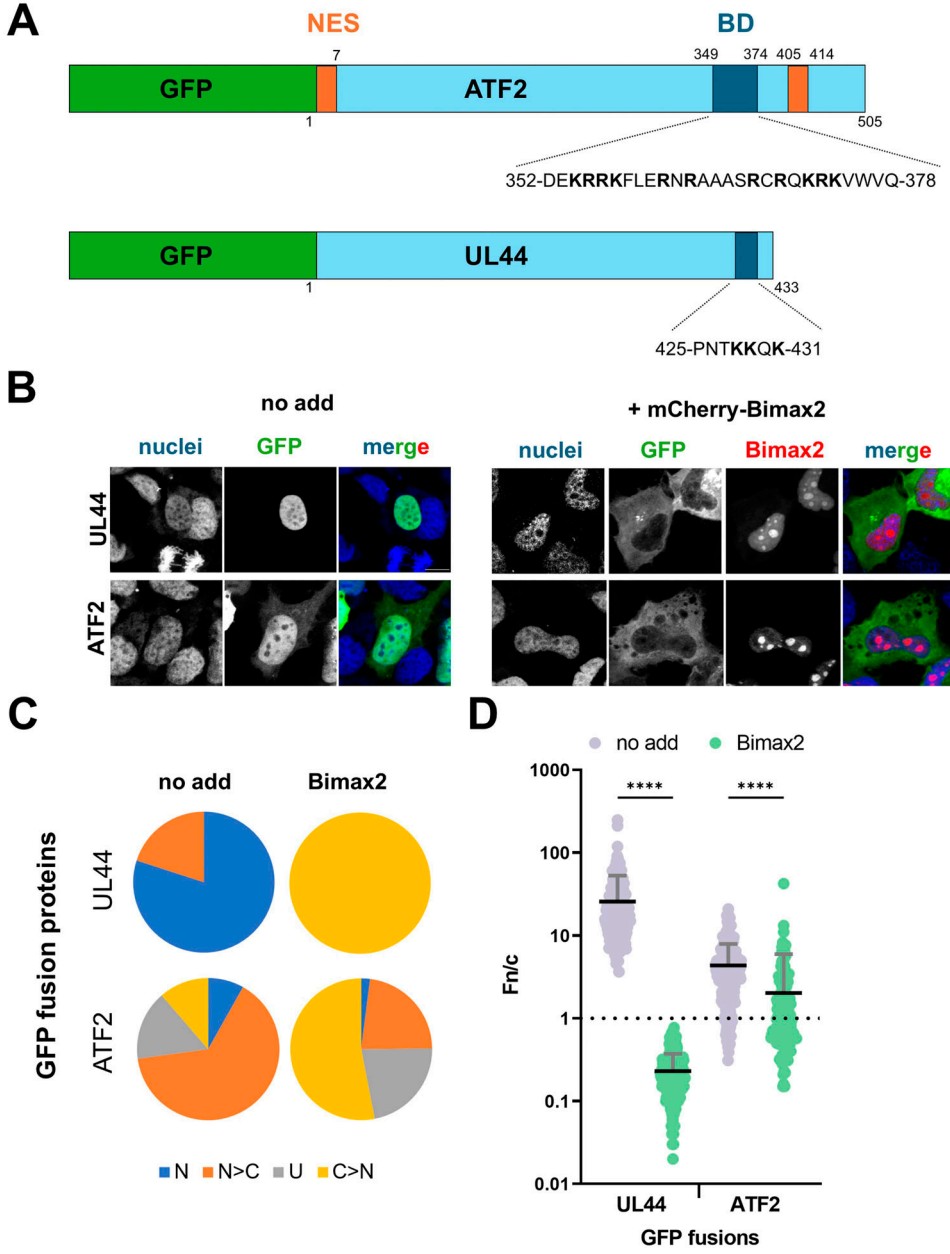

**Figure 3. ATF2 is translocated into the nucleus by the IMPα/β1 heterodimer.**
**(A)** Schematic representation of GFP-fusion protein expression plasmids. Previously described NESs are indicated as orange rectangles. The BD/NLS is in blue. NLS sequences are shown with single-letter amino acids, with basic residues in boldface. NES, nuclear export signal; BD, basic domain; NLS, nuclear localisation signal. **(B)** HEK293A cells were transfected with plasmids encoding the indicated GFP-fusion proteins, in the presence or absence of the IMPα/β1-dependent nuclear transport inhibitor Bimax2. At 24 h post-transfection, cells were stained with DRAQ5 to visualise nuclei, then fixed and analysed by CLSM to quantify nuclear accumulation at the single-cell level. Representative images of the indicated GFP-fusion proteins expressed in the absence (left panels, no addition) or presence (right panels, +mCherry-Bimax2) of mCherry-Bimax2. Images for each channel are shown, along with merged images. Scale bar = 10 μM. **(B, C)** Micrographs, such as those shown in panel (B), were quantitatively analysed to calculate the nuclear-to-cytoplasmic fluorescence ratio (Fn/c) for each GFP-fusion protein at the single-cell level. The percentage of cells showing each category of subcellular localisation is presented: N (nuclear), Fn/c ≥ 10; N > C (nuclear more than cytoplasmic), 2 ≤ Fn/c < 10; U (ubiquitous), 1 ≤ Fn/c < 2; C > N (cytoplasmic more than nuclear), Fn/c < 1. **(D)** Individual Fn/c measurements are shown, along with the mean (black horizontal bars) and SD (grey vertical bars) from three independent experiments. Statistical significance was assessed using Welch's and Brown–Forsythe ANOVA tests; ****$P < 0.0001$.

impaired nuclear import (Figs 6A–C and S4). However, co-expression with RFP-c-Jun significantly increased the nuclear accumulation of both ATF2 ΔBD and ATF2 S1S2m, but not of UL44 ΔNLS. For ATF2 ΔBD, co-expression with RFP-c-Jun increased the mean Fn/c to 1.25, representing a modest but significant shift ($P < 0.0001$). Notably, the effect was far more pronounced for the ATF2 S1S2m mutant, which showed a dramatic increase in nuclear accumulation upon co-expression with RFP-c-Jun (Fn/c increased from 0.34 to 13.10; $P < 0.0001$). These data suggest that ATF2 can be co-transported into the nucleus with c-Jun. However, previous studies demonstrated increased nuclear localisation of ATF2 in the presence of c-Jun because of impairment of nuclear export (2). Therefore, the increased nuclear localisation of ATF2 NLS mutants in the presence of c-Jun could be the consequence of either co-transport in the nucleus with c-Jun, or decreased nuclear export. To discriminate between these two possibilities, we analysed the subcellular localisation of ATF2 S1S2m and ΔBD constructs in the presence of LMB (Figs 6A–C and S4). Intriguingly, inhibition of CRM-1 did not increase nuclear localisation of any protein tested, indicating that their increased nuclear localisation in the presence of

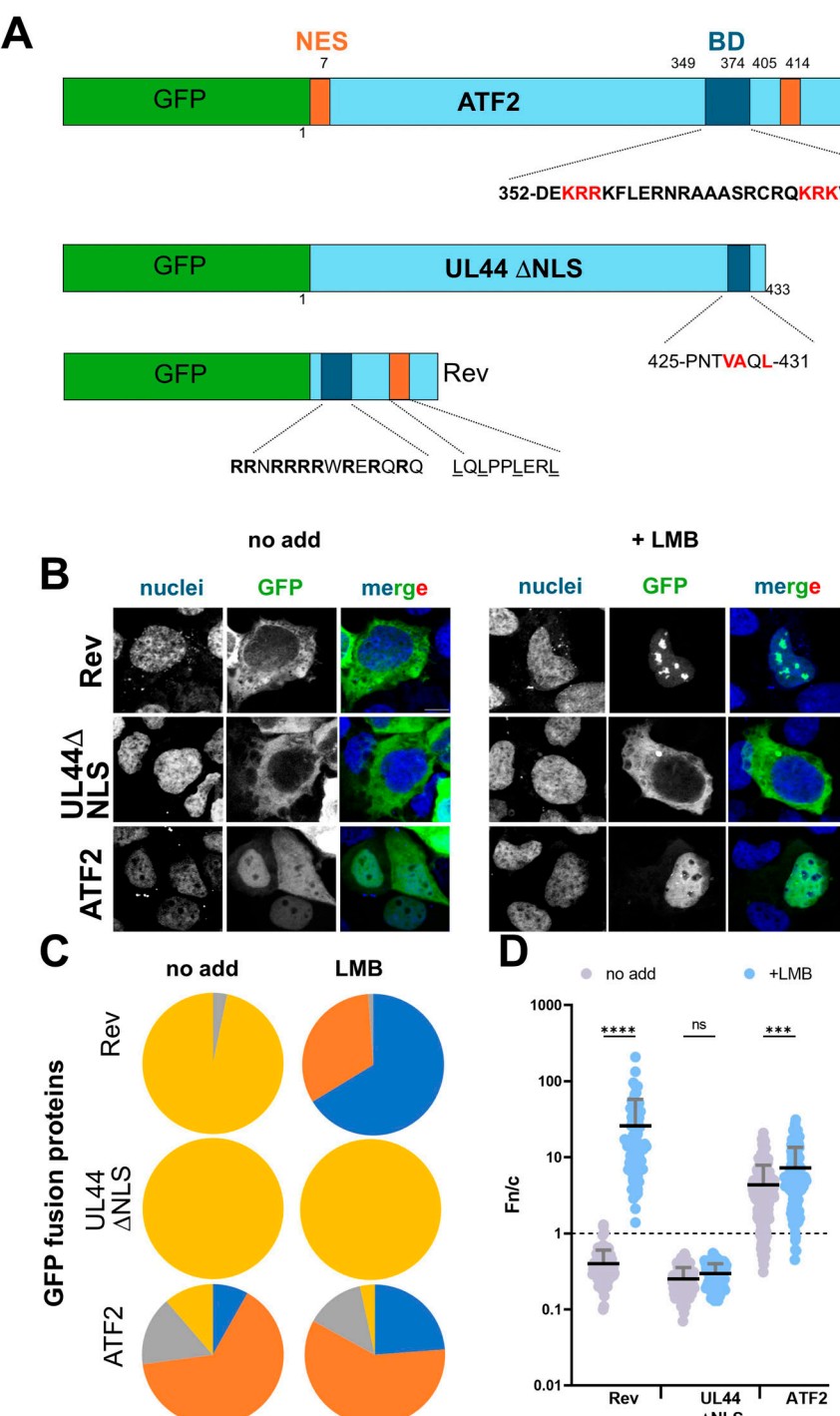

**Figure 4. ATF2 is exported from the nucleus via the CRM1-dependent pathway.**
**(A)** Schematic representation of GFP-fusion protein expression plasmids. Previously described NESs are indicated as orange rectangles. The BD/NLS is in blue. NLS sequences are shown with single-letter amino acids, with basic residues in boldface. HIV-1 Rev NLS sequence is also shown, with hydrophobic residues underlined. NES, nuclear export signal; BD, basic domain; NLS, nuclear localisation signal. **(B)** HEK293A cells were transfected with plasmids encoding the indicated GFP-fusion proteins. At 6 h post-transfection, the media containing DNA–Lipofectamine complexes were replaced with fresh DMEM, either alone (no addition) or supplemented with leptomycin B (LMB; 2.9 ng/ml). At 24 h post-transfection, cells were stained with DRAQ5 to visualise nuclei, fixed, and analysed by CLSM to quantify nuclear accumulation at the single-cell level. Representative images of cells expressing the indicated GFP-fusion proteins in the absence (left panels; no addition) or presence (right panels; + LMB) of LMB. Individual fluorescence channels and merged images are shown. Scale bar = 10 $\mu$M. **(C)** Micrographs were quantitatively analysed to calculate the nuclear-to-cytoplasmic fluorescence ratio (Fn/c) for each GFP-fusion protein. Cells were categorised based on Fn/c values as follows: N (nuclear), Fn/c ≥ 10; N > C (nuclear > cytoplasmic), 2 ≤ Fn/c < 10; U (ubiquitous), 1 ≤ Fn/c < 2; C > N (cytoplasmic > nuclear), Fn/c < 1. **(D)** Individual Fn/c values are shown, along with the mean (black horizontal bars) and SD (grey vertical bars) from three independent experiments. Statistical significance was assessed using Welch's and Brown–Forsythe ANOVA tests; ****$P$ < 0.0001; ***$P$ < 0.001; ns, not significant.

c-Jun is due to co-transport in the nucleus rather than to decreased export. The results support a model in which ATF2 nuclear localisation can occur via two mechanisms: independent nuclear import through IMPα binding and alternative piggyback transport in complex with c-Jun.

## Discussion

Nuclear localisation is a critical step for transcription factors to perform their function of altering nucleosome structure and facilitating transcription. This process is largely dependent on NLSs,

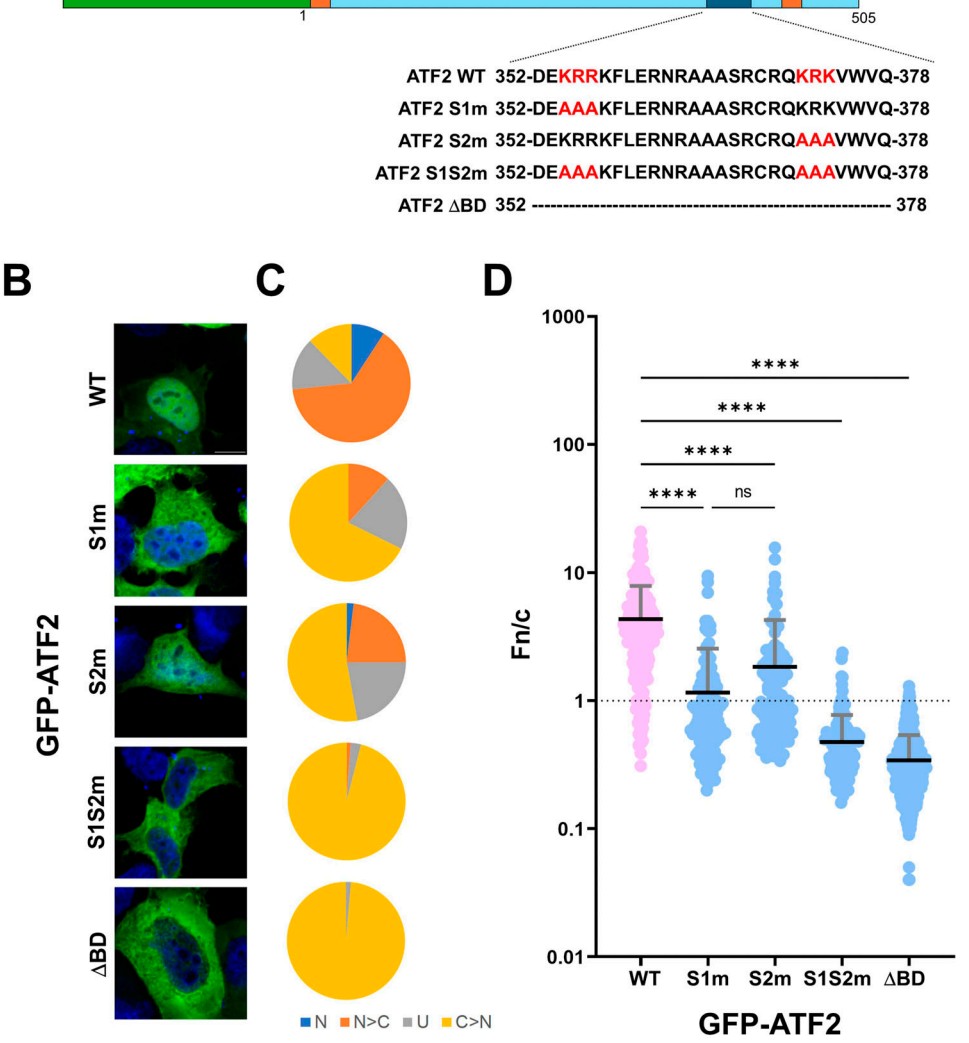

**Figure 5. ATF2 nuclear accumulation requires both basic clusters for efficient import.**
**(A)** Schematic representation of GFP-fusion protein expression plasmids. Previously described NESs are indicated as orange rectangles. The BD/NLS is in blue. NLS sequences are shown with single-letter amino acids, with basic residues in boldface. NES, nuclear export signal; BD, basic domain; NLS, nuclear localisation signal. **(B)** HEK293A cells were transfected with plasmids encoding the indicated GFP-ATF2 constructs. Cells were then stained with DRAQ5 to visualise nuclei, fixed, and analysed by confocal laser scanning microscopy to quantify nuclear accumulation at the single-cell level. Representative images of cells expressing the indicated GFP-fusion proteins. Merged images are shown; DRAQ5 is shown in blue; GFP-ATF2 is shown in green. Scale bar = 10 $\mu$M. **(C)** Micrographs were quantitatively analysed to calculate nuclear-to-cytoplasmic fluorescence intensity ratios (Fn/c) for each GFP construct. Cells were categorised by localisation pattern as follows: N (nuclear), Fn/c ≥ 10; N > C (nuclear more than cytoplasmic), 2 ≤ Fn/c < 10; U (ubiquitous), 1 ≤ Fn/c < 2; C > N (cytoplasmic more than nuclear), Fn/c < 1. **(D)** Individual Fn/c values are shown, along with the mean (black horizontal bars) and SD (grey vertical bars) from three independent experiments. Statistical significance was assessed using Welch's and Brown–Forsythe ANOVA; ****$P$ < 0.0001.

which mediate transport of these proteins into the nucleus. Mutations in NLS regions can impair nuclear import, leading to reduced transcriptional activity and, in some cases, disease. Several clinically documented disorders are linked to such mutations, including Job's syndrome, caused by mutations in or near the NLS of STAT3, which impair its nuclear accumulation (22), Denys–Drash syndrome, associated with mutations in the NLS region of WT1 (23), and Waardenburg syndrome type IV and Hirschsprung disease, both linked to mutations in the NLS of SOX10 (24).

ATF2 is a stress-activated component of the AP-1 complex and plays a role in DNA repair and cell survival (25). Other AP-1 members also have critical functions in the cell. c-Fos and c-Jun heterodimers promote cell cycle progression (1). c-Jun can also induce the expression of pro-apoptotic genes such as *FasL*; however, in certain cancers AP-1 may support cell survival by promoting anti-apoptotic gene expression (26). In addition, AP-1 regulates the expression of inflammatory cytokines such as IL-2, IL-6, and TNF-$\alpha$, as well as matrix metalloproteinases, in immune and stromal cells, playing a key role in the inflammatory response (27). Many studies indicate that unregulated nucleoplasmic accumulation of AP-1 members can contribute to disease. For example, increased nuclear localisation of c-Jun is shown in colorectal cancer tissue (28), and Fra-1 (FOSL1) is frequently overexpressed and retained in the nucleus where it drives epithelial-to-mesenchymal transition and metastasis (29). In addition, dysregulation of Fos nuclear–cytoplasmic shuttling contributes to aggressive tumour behaviour (30). Currently, the mechanism of ATF2 nuclear localisation remains poorly understood.

In this study, we employed a combination of peptides, the isolated bZIP domain, and full-length ATF2 depending on the experimental approach. For quantitative in vitro binding and fluorescence polarisation assays, we focused on the bZIP domain encompassing the basic region identified in our structural

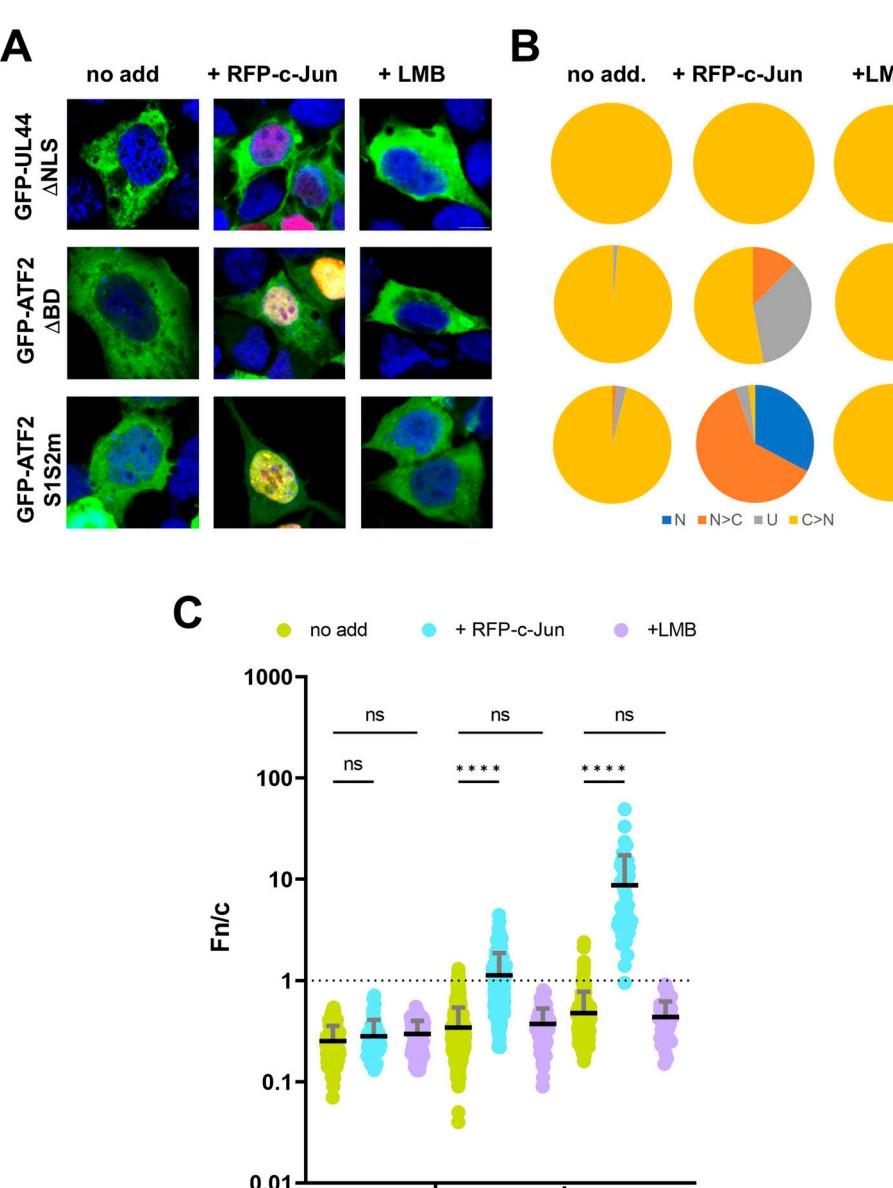

**Figure 6.   ATF2 can be co-transported into the nucleus with c-Jun.**

**(A)** HEK293A cells were transfected with plasmids encoding GFP-UL44ΔNLS, GFP-ATF2ΔBD, or GFP-ATF2 S1S2m, either alone (no addition) or in combination with RFP-c-Jun (+RFP-c-Jun). At 6 h post-transfection, the media containing DNA–lipofectamine complexes were replaced with fresh DMEM, either alone or supplemented with LMB (+LMB). At 24 h post-transfection, cells were stained with DRAQ5 to visualise nuclei, then fixed, and analysed by confocal laser scanning microscopy to quantify nuclear accumulation at the single-cell level. Representative images of each GFP-tagged fusion protein are shown; DRAQ5 is shown in blue; GFP-tagged fusions are shown in green; RFP-c-Jun is shown in red. Scale bar = 10 μM. **(B)** Micrographs were quantitatively analysed to calculate nuclear-to-cytoplasmic fluorescence intensity ratios (Fn/c) for each GFP construct. Cells were categorised by localisation pattern as follows: N (nuclear), Fn/c ≥ 10; N > C (nuclear more than cytoplasmic), 2 ≤ Fn/c < 10; U (ubiquitous), 1 ≤ Fn/c < 2; C > N (cytoplasmic more than nuclear), Fn/c < 1. **(C)** Individual Fn/c values are shown, along with the mean (black horizontal bars) and SD (grey vertical bars) from three independent experiments. Statistical significance was assessed using Welch's and Brown–Forsythe ANOVA; ****P < 0.0001; ns, not significant.

analyses as the principal importin interaction site. This strategy was driven by both technical and mechanistic considerations. Expression and purification of discrete structured domains in *Escherichia coli* are considerably more robust than for large proteins containing extended intrinsically disordered regions, and the bZIP domain retains the core nuclear transport determinants while avoiding solubility and stability limitations associated with the full-length protein. Functionally, the ATF2 bZIP domain is highly enriched in basic residues and contains the DNA-binding and dimerisation elements of ATF2, features that closely resemble classical nuclear localisation signals recognised by importin-α family members. Sequence conservation and prior literature further support this region as the primary nuclear

targeting module of ATF2. Importantly, our cellular assays using full-length ATF2 confirm that these same elements govern nuclear accumulation, validating the physiological relevance of the domain-based approach. Restricting the analysis to the bZIP domain also enabled direct interrogation of protein–protein interactions with importin-α paralogues and importin-β1 in isolation, without confounding contributions from flexible regulatory regions that could influence binding indirectly. Although we cannot exclude the possibility that additional karyopherins may engage ATF2 in specific cellular contexts, we focused on importin-α/β1 because this pathway represents the canonical and best-characterised nuclear import route for basic NLS-containing cargoes. The functional importance of this pathway

in ATF2 nuclear transport is further supported by our BiMax inhibitor experiments in cells. Together, this multiscale strategy allowed us to define the core molecular determinants of ATF2 nuclear import while maintaining clear relevance to full-length protein behaviour in cells.

Our crystal structure of the ATF2 BD in complex with IMPα2, together with supporting quantitative FP-binding data, reveals that ATF2 interacts with IMPα via two distinct basic clusters. These two clusters, [353]EKRRK[359] (site 1) and [372]KRK[374] (site 2), bind to the major and minor sites of IMPα, respectively. However, unlike classical bipartite IMPα–cargo complexes, where NLSs typically span from the N terminus to the C terminus in a minor-to-major binding site orientation, ATF2 exhibits a reversed or inverted binding mode. In canonical bipartite NLSs, the N-terminal basic cluster binds the IMPα minor site (ARM repeats 6-8) and the C-terminal basic cluster binds the IMPα major site (ARM repeats 2–4), as seen in well-characterised complexes such as nucleoplasmin-IMPα, SOX2, and MERS-CoV ORF4b (14, 15, 31). In contrast, ATF2 site 2 binds the minor site, whereas the upstream site 1 binds the major site. This reverse orientation is unusual and suggests that the two basic clusters do not originate from a continuous linear bipartite NLS. Instead, the observed configuration likely reflects contributions from two separate ATF2 molecules. Given that ATF2 functions as a homodimer via its leucine zipper domain, it is plausible that each monomer contributes one basic cluster to the interface, collectively mimicking a "bipartite-like" NLS in trans. This dimeric arrangement would allow simultaneous engagement of both the major and minor sites on a single IMPα molecule, consistent with the high-affinity binding observed in FP assays. Nevertheless, we acknowledge that simultaneous occupancy of both the major and minor binding sites might be due to crystallisation of an isolated ATF2-NLS peptide rather than the binding mode of full-length ATF2 in solution, where steric constraints may occlude the second site.

The previously reported ATF2 bZIP structure (PDB 1T2K) indicates that the basic region adopts an α-helical conformation in the absence of binding partners. Notably, our IMPα2:ATF2 BD crystal structure captures this region in an extended, non-helical conformation, suggesting that local structural rearrangement may accompany IMPα2 engagement. This behaviour is consistent with a broader observation that IMPα paralogues preferentially interact with flexible or unfolded basic regions rather than rigid secondary structures (13, 14, 15, 32, 33). But in some cases, binding to IMPα is associated with helix-to-extended transitions, most prominently illustrated by the IMPα importin-β–binding domain, which undergoes a pronounced conformational change during auto-inhibition and release (34). AlphaFold predictions were inconsistent in modelling interactions between different ATF2 constructs and IMPα1 (Fig S5); however, models generated using the ATF2 BD domain indicate that the predicted conformations are compatible with IMPα binding and support the conformational rearrangement inferred from the peptide complex. Together, these observations suggest that although the ATF2 NLS region adopts a helical conformation in isolation, its conformational plasticity likely enables productive engagement with importin-α during nuclear import.

Our pull-down assay data indicate that ATF2 preferentially interacts with IMPα over IMPβ1, supporting its use of the classical IMPα/β1-mediated nuclear import pathway. This mode of nuclear entry is consistent with that observed for several transcription factors, including SOX2 (14), STAT1 (35, 36), and c-MYC (35), all of which are known to rely on IMPα for nuclear translocation, but in stark contrast to c-Jun, which is transported into the nucleus by multiple nuclear transport receptors alternative to IMPα/β1 (37). Furthermore, our quantitative FP assays revealed that the NLS region of ATF2 exhibits the highest binding affinity for IMPα1, with lower affinities for other paralogues such as IMPα3, IMPα5, and IMPα7. This apparent paralogue preference aligns with previous reports highlighting cargo-specific importin selectivity. For instance, SOX2 shows a marked preference for IMPα3 (14), whereas the Ebola virus protein VP24 selectively binds IMPα5 (11). Such selectivity may be functionally relevant, contributing to tissue-specific nuclear import, differential cargo regulation, or competitive binding in the cellular context. To experimentally demonstrate that ATF2 uses the classical nuclear import pathway, we used the well-known IMPα inhibitor Bimax2, which has been shown to inhibit nuclear import of several IMPα/β1-dependent cargoes (38, 39, 40), but not of histone H1E (41) or adenovirus pVII proteins (18, 19, 42), which use alternative nuclear import pathways. Treatment of cells with Bimax2 significantly reduced the nuclear localisation of ATF2, indicating that ATF2 relies on the IMPα/β1 pathway to enter the nucleus. However, unlike our positive control protein UL44, ~25% of the cells still showed ATF2 predominantly in the nucleus after Bimax2 treatment, suggesting that ATF2 may also employ alternative pathways for nuclear entry.

A previous study by Liu et al (2) proposed that ATF2 contains a classical bipartite NLS comprising two basic motifs, [342]RRRR[345] (referred to as NLS1) and [354]KRRK[357] (referred to as NLS2). Our data support a slightly different model. Through protein crystallisation, FP-binding assays, and cellular nuclear localisation assays, our findings identify the functional NLS motifs as [354]KRR[356], consistent with NLS2 described by Liu et al (2), and [372]KRK[374], which lies downstream of the previously proposed NLS1. Consistent with our FP-binding data using ATF2 site mutants, nuclear localisation assays demonstrated that mutation of site 1 had a more pronounced effect on nuclear accumulation than mutation of site 2. Furthermore, simultaneous mutation of both motifs completely abolished nuclear localisation, indicating that both sites are required for efficient nuclear import. These observations were further substantiated by LMB inhibition assays, which assess CRM1-dependent nuclear export. Upon LMB treatment, ATF2 S1S2m nuclear accumulation was not increased, confirming that in the absence of both NLS motifs, nuclear entry is completely abolished. The differences between our results and those of Liu et al (2) may be attributed to several factors. First, the two studies used different cell lines, which may vary in importin paralogue expression, nuclear transport dynamics, or transcription factor profiles that influence ATF2 localisation. Second, ATF2 is known to heterodimerise with other bZIP transcription factors, including Jun family members, which could facilitate nuclear import through piggyback mechanisms. Such non-classical or cooperative transport pathways may explain the residual nuclear localisation observed in the Liu et al (2) study, even in the absence of both proposed NLS motifs.

Nuclear localisation through heterodimerisation has been reported in previous studies. For instance, one study demonstrated

that c-Jun can enter the nucleus by forming a heterodimer with cFOS (37). In our studies using RFP-tagged c-Jun and ATF2 mutants incapable of being transported into the nucleus on their own, we observed a significant increase in nuclear entry after co-transfection with c-Jun. The increase in nuclear localisation was more pronounced for ATF2 S1S2m compared with ATF2 ΔBD, which may be due to a structural deficiency in the bZIP domain of ATF2 ΔBD, reducing its ability to heterodimerise efficiently with c-Jun. These findings are consistent with a previous report showing increased nuclear localisation of ATF2 in the presence of c-Jun, but are partially in contrast to the hypothesis that such increase is due to a reduction in CRM1-dependent nuclear export (2). Indeed, our data clearly show that both ATF2 S1S2m and ATF2 ΔBD are not exported from the nucleus by CRM1 because of abolished nuclear import.

Despite the evidence presented in this study, further investigation is required to fully elucidate the complex mechanisms governing ATF2 nuclear localisation. Although our structural and functional data strongly support the role of two basic clusters in mediating import via the classical IMP$\alpha$/$\beta$1 pathway, with site 1 acting as the dominant motif for import, observations from our cellular assays suggest that additional pathways may contribute to ATF2 nuclear translocation. These may include import mediated by non-classical importins, or alternative mechanisms such as piggybacking through heterodimerisation with other AP-1 family members. Such alternative routes could be context-dependent and influenced by cell type, signalling environment, or post-translational modifications. Further studies, including IMP$\alpha$ paralogue–specific knockdowns, interaction mapping, and live-cell imaging, will be essential to dissect these mechanisms in greater detail. Nevertheless, the present study provides a robust structural and biochemical framework for understanding ATF2 nuclear import and offers a solid foundation for exploring the broader regulatory landscape of ATF2 trafficking and function.

## Materials and Methods

### Gene construct design

The amino acid sequence corresponding to the bZIP domain of human ATF2 (residues 350–417; sequence: DPDEKRRKFLERNRAAAS RCRQKRKVWVQSLEKK AEDLSSLNGQLQSEVTLLRNEVAQLKQLLLAHKD), hereafter referred to as ATF2 bZIP, was retrieved from the GenBank database (accession number X15875.1). The nucleotide sequence encoding this domain was codon-optimised for expression in *E. coli*. The synthetic gene included an N-terminal tobacco etch virus (TEV) protease recognition site to enable subsequent removal of a His-tag and was cloned into the pET30a(+) expression vector via *BamHI* restriction sites (GenScript).

For biochemical and structural studies, ΔIBB (importin-$\beta$–binding domain–deleted) constructs of human IMP$\alpha$ proteins were generated. These included human IMP$\alpha$1 (hIMP$\alpha$1, NM_001320611.3, residues 71–529), hIMP$\alpha$3 (NM_002268.5, residues 64–521), hIMP$\alpha$5 (XM_005247437.5, residues 74–538), hIMP$\alpha$7

(NM_012316.5, residues 74–536), and mouse IMP$\alpha$1 (also known as IMP$\alpha$2, NM_010655.3, residues 71–529). All coding sequences were codon-optimised for *E. coli* expression and synthesised by Gen-Script. All constructs contained an N-terminal His$_6$ affinity tag. All constructs, except IMP$\alpha$2, in addition contained an N-terminal TEV protease cleavage site to allow removal of the affinity tag. IMP$\alpha$2 was cloned into the pET15b expression vector, whereas all other IMP$\alpha$ constructs were cloned into the pET30a(+) vector via the BamHI restriction site.

Full-length human IMP$\beta$1 (IMP$\beta$1, Q14974, residues 1–876) was cloned into the pMCSG21 expression vector with an N-terminal His$_6$ affinity tag followed by a TEV protease cleavage site for tag removal.

For cellular assays, the open reading frames (ORFs) of full-length human *ATF2* and *JUN* genes (GenBank accession numbers X15875.1 and NM_002228.4, respectively) were cloned into the pcDNA3.1(+) expression vector with an N-terminal GFP tag using Gateway-compatible attB1 and attB2 recombination sites flanking the gene. Four ATF2 mutant constructs were generated using the same cloning strategy: (1) ATF2 site 1 mutated (S1m): residues $^{354}$KRR$^{356}$ mutated to AAA; (2) ATF2 site 2 mutated (S2m): residues $^{372}$KRK$^{374}$ mutated to AAA; (3) ATF2 site 1 and site 2 double mutant (S1S2m): both site 1 ($^{354}$KRR$^{356}$ to AAA) and site 2 ($^{372}$KRK$^{374}$ to AAA) mutated; and (4) ATF2 ΔBD: a deletion mutant lacking the entire BD (residues 348–377). The ORF of *JUN* was transferred to expression plasmid pDEST-ntRFP by means of LR Gateway recombination (17). Plasmid pEPI-GFP-UL44, mediating the expression of the DNA polymerase processivity factor from human cytomegalovirus, which is transported into the nucleus by the IMP$\alpha$/$\beta$1 heterodimer, and its NLS defective derivative pEPI-GFP-UL44ΔNLS were described previously (21). Plasmid pEGFP-Rev, mediating the expression of human immunodeficiency virus type 1 (HIV-1) Rev protein (43), which is exported from the nucleus by chromosomal region maintenance 1 (CRM1), was provided by Reena Ghildyal (Canberra, Australia), whereas pCDNA3-mCherry-Bimax2, encoding a competitive inhibitor of the IMP$\alpha$/$\beta$1 nuclear import pathway (20, 44), was kindly gifted from Yoshihiro Yoneda and Masahiro Oka (Osaka, Japan).

### Fluorescein isothiocyanate (FITC)–labelled peptide design

Four N-terminal FITC-tagged Ahx-linker peptides with BD sequences were designed for this study, including ATF2 WT (352-DE**KRR**KFLERNRAAASRCRQ**KRK**VWVQ-378), ATF2 S1m (352-DE**AAA**KFLERNRAAASRCRQ**KRK**VWVQ-378), ATF2 S2m (352-DE**KRR** KFLERNRAAASRCRQ**AAA**VWVQ-378), and ATF2 S1S2m (352-DE**AAA** KFLERNRAAASRCRQ**AAA**VWVQ-378) (site 1 and site 2 residues shown in bold). The peptides were obtained through GenScript.

### Protein expression and purification

Importin paralogue and ATF2 bZIP were overexpressed in *E. coli* BL21(DE3) pLysS cells using the autoinduction method as previously described (45, 46). After induction, cells were harvested by centrifugation at 6,400 RCF for 20 min and resuspended in 20 ml of His buffer A (50 mM sodium phosphate, 300 mM NaCl, 20 mM imidazole, pH 8.0) per 2 litre culture volume. Cell lysis was

performed by two cycles of freezing and thawing, followed by treatment with 1 ml of 20 mg/ml lysozyme (Sigma-Aldrich) and 10 $\mu$l of 50 mg/ml DNase I (Sigma-Aldrich) at RT for 1 h. The clear supernatant from the lysate was achieved by centrifugation at 11,800 RCF for 30 min, and the supernatant was passed through a 0.45-$\mu$m low protein-binding filter. The supernatant containing the overexpressed protein was then injected onto a 5-ml HisTrap HP affinity column (GE Healthcare) connected to an AKTA Pure FPLC system (GE Healthcare), followed by 20 column volumes of wash by His buffer A, before the expressed proteins were eluted using a linear imidazole gradient ranging from 20 to 500 mM. Elution fractions containing the target proteins were pooled and subjected to further purification via size-exclusion chromatography using a HiLoad 26/600 Superdex 200 column (GE Healthcare) pre-equilibrated in GST buffer A (50 mM Tris–HCl, 125 mM NaCl, pH 8.0). Relevant fractions, identified based on expected elution volume and molecular weight, were concentrated using 10 kD molecular weight cut-off Amicon centrifugal filters (Merck Millipore) and stored in aliquots at −80°C. Protein purity was confirmed by SDS–PAGE, run at 165 V for 30 min on 4–12% Bis-Tris Plus gels (Thermo Fisher Scientific).

### Nickel bead affinity pull-down assay

In this binding assay, 10 $\mu$M of His-tagged target protein (ATF2 bZIP) was incubated with 50 $\mu$l of Ni-NTA His-bind resin beads (Cat. No. 70666; Millipore) in a total volume of 300 $\mu$l using His buffer A. To this mixture, 20 $\mu$M of the interacting proteins (IMP paralogue) was added. Before washing, a 30 $\mu$l aliquot of the reaction was combined with 30 $\mu$l of SDS sample buffer to generate input samples. The resin was subsequently washed three times with His buffer A to remove unbound proteins. The complex was eluted using 10 $\mu$g TEV enzyme. From each input and output preparation, 20 $\mu$l was loaded onto a 4–12% Bis-Tris SDS–PAGE gel (Thermo Fisher Scientific) and electrophoresed at 165 V and 120 mA for 30 min. Gels were stained with Coomassie brilliant blue and destained according to standard protocols. Final gel images were captured using the GelDoc system and analysed using ImageLab software (version 6.0.1) (15).

### Fluorescence polarisation (FP) assays

FP assays were employed to quantitatively assess binding affinities between ATF2 peptides and IMP proteins. To perform the assay, a twofold serial dilution of IMP protein was prepared across 23 wells of a black 96-well microplate, with the 24th well containing only assay buffer (50 mM Tris [Cat. No. TR0425005P; CSA Scientific], pH 8.0, 125 mM NaCl). A fixed concentration of the fluorescently labelled ATF2 peptide was then added to all wells, to reach a final concentration of 5 nM. Well 24 served as a background control containing only the peptide and buffer. FP measurements (three replicates) were recorded by the CLARIOstar Plus system (BMG Labtech), and resulting data were analysed using non-linear regression with one site-specific binding in GraphPad Prism software (version 9.4) (46).

### Crystallisation of ATF2 BD peptide and IMP$\alpha$2 complex

The IMP$\alpha$2:ATF2 BD complex was prepared by mixing the two components at a 1:2 M ratio. Crystallisation was performed using the hanging-drop vapour diffusion method at 23°C. The final crystallisation condition consisted of 0.7 M sodium citrate, 0.1 M Hepes (pH 7.0), and 10 mM DTT, mixed in a 1:1 ratio with the protein complex solution (resulting in a final protein concentration of 5 mg/ml). Rod-shaped crystals appeared within 3–5 d. Crystals were harvested and cryoprotected using reservoir solution supplemented with 20% (vol/vol) glycerol, then flash-cooled in liquid nitrogen for data collection.

### Data collection and structure determination

X-ray diffraction data were collected at the Australian Synchrotron on the MX2 beamline using an Eiger 16 M detector (47). The integrated file was downloaded from the Synchrotron, and scaling and merging were performed using Aimless (48). The structure was solved by molecular replacement using Phaser (49), with PDB entry 4OIH serving as the search model. Iterative model building and refinement were performed using Coot (50) and Phenix (51), respectively.

### Cell culture and transfection

HEK293A cells were maintained in complete growth medium consisting of DMEM supplemented with 10% foetal bovine serum, 50 U/ml each of penicillin and streptomycin, and 2 mM L-glutamine. Cultures were incubated at 37°C in a humidified atmosphere containing 5% $CO_2$ and routinely subcultured once they reached confluency. For transfection experiments, cells were plated onto sterile glass coverslips placed in 24-well plates at a density of 5 × $10^4$ cells per well. After 24 h, cells were transfected with 5–250 ng of plasmid DNA using Lipofectamine 2000 (Thermo Fisher Scientific), following the manufacturer's protocol. Post-transfection, cells were incubated under standard culture conditions in complete medium until further processing (40).

### Confocal laser scanning microscopy (CLSM) and quantitative image analysis

Twenty-four hours after transfection, cells were incubated with DRAQ5 nuclear stain (Cat. No. 62251; 1:5,000 dilution in phenol red–free DMEM; Thermo Fisher Scientific) for 30 min. After staining, cells were washed twice with 1× PHEM buffer (60 mM PIPES, 25 mM Hepes, 10 mM EGTA, 4 mM $MgSO_4$) and subsequently fixed with 4% PFA in PHEM buffer for 10 min at RT. After fixation, cells were rinsed three times with 1× PBS and mounted on microscope slides using Fluoromount-G (SouthernBiotech). Protein subcellular localisation was examined using a Nikon A1 confocal laser scanning microscope (Nikon) fitted with a 60× oil-immersion Plan-Apochromatic objective (Nikon Plan Apol lambda, 1.40 N.A., and 0.13 M.M. working distance, Cat. No. MRD01605), as described in reference 19. Levels of nuclear accumulation of proteins of interest were determined using Fiji public domain software (52) from single-cell measurements of both the nuclear (Fn) and cytoplasmic (Fc) fluorescence,

subsequent to the subtraction of fluorescence because of autofluorescence/background, as described previously. In indicated cases (53), leptomycin B (LMB, Cat. No. L2913; 2.9 ng/ml; Sigma-Aldrich) was added to cells 6 h before processing samples for imaging, as previously described (54). Data were statistically analysed by performing either a *t* test or one-way ANOVA using GraphPad Prism software (version 9) as appropriate.

## Data Availability

Files associated with the structure generated in this study have been deposited to the Protein Data Bank and were released before submission of the article with PDB ID: 9Y0R. Source data are provided with the article.

## Supplementary Information

## Acknowledgements

This research was undertaken in part using the MX2 beamline at the Australian Synchrotron, part of ANSTO, and made use of the Australian Cancer Research Foundation (ACRF) detector.

### Author Contributions

SM Ghafoori: conceptualisation, data curation, formal analysis, investigation, methodology, and writing—original draft, review, and editing.
S Pavan: formal analysis, investigation, and methodology.
TX Duc: data curation, formal analysis, and methodology.
S Nematollahzadeh: data curation and formal analysis.
GF Petersen: data curation, methodology, and writing—original draft, review, and editing.
G Alvisi: conceptualisation, data curation, formal analysis, supervision, funding acquisition, investigation, visualisation, methodology, project administration, and writing—original draft, review, and editing.
JK Forwood: conceptualisation, data curation, formal analysis, supervision, funding acquisition, validation, investigation, visualisation, methodology, project administration, and writing—original draft, review, and editing.

### Conflict of Interest Statement

The authors declare that they have no conflict of interest.

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
