## [Reviewer comments · Life Science Alliance]

A non-canonical in trans mechanism and paralog-selective recognition governs ATF2 nuclear import.

Seyed Mohammad Ghafoori, Silvia Pavan, Trinh Xuan Duc, Sepehr Nematollahzadeh, Gayle Petersen, Gualtiero Alvisi, and Jade Forwood

DOI: <https://doi.org/10.26508/lsa.202503543>

Corresponding author(s): Jade Forwood, Charles Sturt University

Review Timeline:

Submission Date:	2025-10-22
Editorial Decision:	2026-01-02
Revision Received:	2026-01-21
Accepted:	2026-01-26

Scientific Editor: Sarita Hebbar

Transaction Report:

January 2, 2026

RE: Life Science Alliance Manuscript #LSA-2025-03543-T

Dr. Jade K Forwood
Charles Sturt University
School of Biomedical Sciences
Boorooma St
Wagga Wagga 2650
Australia

Dear Dr. Forwood,

Thank you for submitting your revised manuscript entitled "A non-canonical in trans mechanism and paralog-selective recognition governs ATF2 nuclear import". We apologise for the delay in communicating our decision due to editor availability issues and delays in securing reviewer comments.

The manuscript was assessed by two expert reviewers, whose comments are appended to this letter. As you will note both reviewers are consistent in their appreciation of this study and its quality. That said, both reviewers have raised few questions and concerns that we concur are important to be addressed. These include providing a rationale choice of bZip domain (Rev 2), clarification for design and results of binding assays (both reviewers), a proper interpretation of the crystal structure and contextualisation with full length ATF 2 structure (Rev 2), and inclusion of pertinent references (both reviewers). We also encourage you to address the minor points raised by both reviewers on HIV-1 Rev NLS sequence (Rev 1) and on presentation of results (Rev 2). Finally, if possible, do provide a supplementary figure with the electron density for the ATF2 peptides as suggested by Rev 2.

Given the overall positive evaluation, we would be happy to publish your paper in Life Science Alliance pending resolution of the above-mentioned reviewers' concerns and final changes necessary to meet our formatting guidelines. In your next submission, please include a letter addressing the reviewers' comments point by point and highlighting any changes made to the text.

- Please provide scale bar information in the legend for Figure 3B and Figure 4B.
- Kindly provide details on the synthesis of constructs for biochemical and structural assays (statement starting - line 88).
- Please provide details for objective used (including objective type and numerical aperture) under the description of imaging method.
- Please clarify in the methods if imaging of GFP fusion proteins was done with the help of any antibodies and if so, provide details of antibodies used.
- Please upload all figure files as individual ones, including the supplementary figure files; all figure legends should only appear in the main manuscript file.
- Please conduct a thorough spell and grammar check of the manuscript text.
- Please add a Summary Blurb/Alternate Abstract in our system.
- Please add a Category and Keywords for your manuscript in our system.
- Please add the X and Bluesky handles of your host institute/organization, as well as your own and/or one of the authors, in our system.
- Please consult our manuscript preparation guidelines <https://www.life-science-alliance.org/manuscript-prep> and make sure your manuscript sections are in the correct order.
- Tables can be included at the bottom of the main manuscript file or sent as separate files.
- It is recommended to exclude figures from the manuscript text and upload them separately.
- Please add an Author Contributions section to your main manuscript text and also in the system.
- The "Data Availability" section should be placed after the Materials & Methods section. Please consult our guidelines at <https://www.life-science-alliance.org/manuscript-prep#format>
- Please add the Acknowledgments section and Conflict of Interest statement to your main manuscript text.
- Please add your main, supplementary figure, and table legends to the main manuscript text after the references section;
- Since your figure S2 has only one panel, please remove label A from it.
- Please add callouts for Figure 6A-C to your main manuscript text.
- Please be sure that the authorship listing and order is correct

LSA now encourages authors to provide a 30-60 second video where the study is briefly explained. We will use these videos on

social media to promote the published paper and the presenting author (for examples, see <https://docs.google.com/document/d/1-UWCfbE4pGcDdcgzcmiuJl2XMBJnxKYeqRvLLrLSo8s/edit?usp=sharing>). Corresponding or first-authors are welcome to submit the video. Please submit only one video per manuscript. The video can be emailed to contact@life-science-alliance.org

A. FINAL FILES:

B. MANUSCRIPT ORGANIZATION AND FORMATTING:

Thank you for your attention to these final processing requirements. Please revise and format the manuscript and upload materials as soon as you are able.

Best wishes for a happy new year,

Sarita Hebbar, PhD
Scientific Editor
Life Science Alliance
<http://www.lsjournal.org>

Reviewer #1 (Comments to the Authors (Required)):

In this paper, Ghafoori et al. present a compelling structure-function analysis of ATF2 nuclear import, dissecting the

requirements for importin α 1/ β - and Crm1-dependent nuclear import and export, respectively. The manuscript is exceptionally well written, beautifully illustrated, and appropriately referenced. The data are quantitative and rigorous, accompanied by sound statistical validation. I have no major concerns; this paper truly represents work of the highest quality.

I have only a few very minor comments:

1. Figure 2: It seems somewhat overstretched to provide a K_D value for a binding event that does not clearly reach saturation (e.g., ATF2 S2m). However, I appreciate that the large standard deviation appropriately informs the reader that the measurement is intrinsically imprecise due to the low binding affinity.
2. Figure 4A: The HIV-1 Rev NLS sequence shown is close to what has been reported in the literature but has never been fully characterized at the molecular level. The actual sequence may be somewhat longer.
3. Reference 45: please also cite Fagerlund et al (PMID: 12048190), who also reported the need for importin α 5 in the STAT1 import pathway

Overall, this is an outstanding manuscript, and I strongly recommend it for immediate publication.

Reviewer #2 (Comments to the Authors (Required)):

The authors conducted an extensive analysis of the interactions between the bZIP domain of the transcription factor ATF2 and the nuclear import receptor Importin- α . Using binding assays and fluorescence polarization measurements, they identify Imp α 1 as the Importin with the highest affinity for the NLS-containing region of ATF2. The crystal structure of the Imp α 1-ATF2_NLS complex reveals two distinct NLS sites on ATF2 that bind independently to the major and minor binding pockets of Imp α , rather than forming a canonical bipartite NLS. Mutational analyses show that residues engaging in the major binding pocket are essential for Imp α 1 binding, but not for binding to other Importins. Finally, in vitro assays using BIMAX, an Imp α /Imp β inhibitor demonstrates that full-length ATF2 predominantly relies on the Imp α / β pathway for nuclear import. Overall, the results are clear, the manuscript is well written, and the findings are compelling. The article should be ready for publication after the authors address the following comments:

- Rationale for using only the bZIP domain.

The introduction or discussion should more clearly explain why only the bZIP domain of ATF2 was used for binding assays and fluorescence polarization measurements. Are there additional regions of ATF2 known or predicted to interact with Importin- α ? Could other karyopherins bind to the same region? Why were only Importins- α and β 1 evaluated?

- Interpretation of the crystal structure and stoichiometry.

In the crystal structure, the ATF2-NLS peptide occupies both the major and minor binding pockets on Imp α 1, suggesting two ATF2-NLS peptides per Imp α 1 molecule. The authors propose that ATF2 may bind Imp α 1 as a homodimer, with each monomer engaging in a different pocket. Because this hypothesis is not experimentally tested, the authors should acknowledge that binding at the minor site may be an artifact of crystallizing isolated short peptides. In the context of the full-length protein, the second site may be occluded, resulting in exclusive binding at the major site.

- Contextualization with the full-length ATF2 structure.

The region containing the putative ATF2 NLSs forms an alpha-helix in both the AlphaFold model (AF-P15336-F1-v6) and the PDB structure 1T2K. The authors should discuss how these model fits with the Imp α 1-ATF2_NLS crystal structure. One possible experiment would be an in-silico analysis using the AlphaFold Server to model binding of Imp α 1 to various ATF2 constructs (e.g., monomer, dimer, isolated NLS region). Summarizing the results in a main or supplementary figure would substantially strengthen the discussion.

Minor comments:

Figure 1B: Show the full gel and label His-ATF2 to clarify which protein is immobilized.

Figures 1C and 2: Clearly indicate in every panel which molecule is fluorescently labeled and specify the fluorophore used.

Figure 1D: Label the N- and C-termini of Imp α and ATF2, and include labels for the ARM repeats.

Figures 3C, 4C, 5B, and 6B: Consider replacing pie charts with bar charts to improve visualization of ratio data.

Include a supplementary figure showing the electron density for the ATF2 peptides in the crystal structures.

Dear Editor

We are grateful to the Editor and the reviewers for their insightful and constructive comments. We have carefully considered all suggestions and revised the manuscript accordingly. Below, we provide a point-by-point response to each comment, with changes highlighted in the revised manuscript.

Reviewer 1:

Comment 1: Figure 2: It seems somewhat overstretched to provide a K_D value for a binding event that does not clearly reach saturation (e.g., ATF2 S2m). However, I appreciate that the large standard deviation appropriately informs the reader that the measurement is intrinsically imprecise due to the low binding affinity.

Response: We agree with the reviewer that for several low-affinity interactions, including mutations designed to affect interaction, the binding curves do not fully reach saturation, which limits the precision of the derived values. We have therefore interpreted these affinities with appropriate caution. As noted, the large standard deviations explicitly reflect this intrinsic uncertainty and are intended to transparently convey the limitations of the measurement rather than imply high confidence in the absolute values. Importantly, our intention in reporting these values is comparative rather than absolute, to illustrate the substantial loss of binding caused by these mutations relative to the wild-type interaction. This approach is consistent with common practice in the field when characterising weak or transient interactions. We have clarified this point in the revised manuscript to ensure the limitations of these measurements are explicitly acknowledged. The figure 2 legend has the following addition:

For low-affinity mutant interactions where saturation was not fully achieved, the precision of the derived K_D values is low, and these values are intended primarily for comparative assessment of relative binding losses to the WT rather than as precise absolute values.

Comment 2: Figure 4A: The HIV-1 Rev NLS sequence shown is close to what has been reported in the literature but has never been fully characterized at the molecular level. The actual sequence may be somewhat longer.

Response: We agree that the full sequence may be longer; however, the purpose of showing the NLS sequence was to highlight the arginine- and leucine-rich features of the motif, rather than to imply that this represents the complete or exact NLS.

Comment 3: Reference 45: please also cite Fagerlund et al (PMID: 12048190), who also reported the need for importin $\alpha 5$ in the STAT1 import pathway.

Response: The mentioned reference is added as requested by the reviewer.

Reviewer 2:

Comment 1: Rationale for using only the bZIP domain. The introduction or discussion should more clearly explain why only the bZIP domain of ATF2 was used for binding assays and fluorescence polarization measurements. Are there additional regions of ATF2 known or predicted to interact with Importin- α ? Could other karyopherins bind to the same region? Why were only Importins- α and β 1 evaluated?

Response: We thank the reviewer for these comments. To address this point, the following paragraph has been added to the Discussion section:

“In this study, we employed a combination of peptides, the isolated bZIP domain, and full-length ATF2 depending on the experimental approach. For quantitative *in vitro* binding and fluorescence polarisation assays, we focused on the bZIP domain encompassing the basic region identified in our structural analyses as the principal importin-interaction site. This strategy was driven by both technical and mechanistic considerations. Expression and purification of discrete structured domains in *E. coli* is considerably more robust than for large proteins containing extended intrinsically disordered regions, and the bZIP domain retains the core nuclear transport determinants while avoiding solubility and stability limitations associated with the full-length protein. Functionally, the ATF2 bZIP domain is highly enriched in basic residues and contains the DNA-binding and dimerisation elements of ATF2, features that closely resemble classical nuclear localisation signals recognised by Importin- α family members. Sequence conservation and prior literature further support this region as the primary nuclear targeting module of ATF2. Importantly, our cellular assays using full-length ATF2 confirm that these same elements govern nuclear accumulation, validating the physiological relevance of the domain-based approach. Restricting the analysis to the bZIP domain also enabled direct interrogation of protein–protein interactions with Importin- α paralogs and Importin- β 1 in isolation, without confounding contributions from flexible regulatory regions that could influence binding indirectly. While we cannot exclude the possibility that additional karyopherins may engage ATF2 in specific cellular contexts, we focused on Importin- α / β 1 because this pathway represents the canonical and best-characterised nuclear import route for basic NLS-containing cargos. The functional importance of this pathway in ATF2 nuclear transport is further supported by our BiMax inhibitor experiments in cells. Together, this multi-scale strategy allowed us to define the core molecular determinants of ATF2 nuclear import while maintaining clear relevance to full-length protein behaviour in cells.”

Comment 2: Interpretation of the crystal structure and stoichiometry. In the crystal structure, the ATF2-NLS peptide occupies both the major and minor binding pockets on Imp α 1, suggesting two ATF2-NLS peptides per Imp α 1 molecule. The authors propose that ATF2 may bind Imp α 1 as a homodimer, with each monomer engaging in a different pocket. Because this hypothesis is not experimentally tested, the authors should acknowledge that binding at the minor site may be an artifact of crystallizing isolated short peptides. In the context of the full-length protein, the second site may be occluded, resulting in exclusive binding at the major site.

Response: We appreciate the reviewer’s comment. The following paragraph has been added to the Discussion section to address this point:

“Nevertheless, we acknowledge that simultaneous occupancy of both the major and minor binding sites might be due to crystallisation of an isolated ATF2-NLS peptide rather than the

binding mode of full-length ATF2 in solution, where steric constraints may occlude the second site.”

Comment 3: Contextualization with the full-length ATF2 structure. The region containing the putative ATF2 NLSs forms an alpha-helix in both the AlphaFold model (AF-P15336-F1-v6) and the PDB structure 1T2K. The authors should discuss how these models fits with the Imp α 1-ATF2 NLS crystal structure. One possible experiment would be an in-silico analysis using the AlphaFold Server to model binding of Imp α 1 to various ATF2 constructs (e.g., monomer, dimer, isolated NLS region). Summarizing the results in a main or supplementary figure would substantially strengthen the discussion.

Response: We appreciate the reviewer’s comment. The following paragraph, together with Supplementary Figure 5, has been added to the Discussion section to address this point: “The previously reported ATF2 bZIP structure (PDB 1T2K) indicate that the basic region adopts an α -helical conformation in the absence of binding partners. Notably, our IMP α 2:ATF2 BD crystal structure captures this region in an extended, non-helical conformation, suggesting that local structural rearrangement may accompany IMP α 2 engagement. This behaviour is consistent with a broader observation that IMP α paralogues preferentially interact with flexible or unfolded basic regions rather than rigid secondary structures (13-15, 31, 32). But in some cases, binding to IMP α is associated with helix-to-extended transitions, most prominently illustrated by the IMP α importin- β -binding domain, which undergoes a pronounced conformational change during auto-inhibition and release (33). AlphaFold predictions were inconsistent in modelling interactions between different ATF2 constructs and IMP α 1 (Supplementary Figure 5); however, models generated using the ATF2 BD domain indicate that the predicted conformations are compatible with IMP α binding and support the conformational rearrangement inferred from the peptide complex. Together, these observations suggest that although the ATF2 NLS region adopts a helical conformation in isolation, its conformational plasticity likely enables productive engagement with Importin- α during nuclear import.

Comment 4: Figure 1B: Show the full gel and label His-ATF2 to clarify which protein in immobilized.

The full images of the gels have been added to the supplementary as requested by the reviewer (see supplementary Figure S1).

Comment 5: Figures 1C and 2: Clearly indicate in every panel which molecule is fluorescently labelled and specify the fluorophore used.

Both panels have been edited regarding editor’s comment as below:

Figure 1: Fluorescence polarization assays measuring the binding affinity between FITC-labelled ATF2 BD and IMP α paralogs.

Figure 2: Fluorescence polarization assays measuring the binding affinity between wildtype (WT) and mutant FITC-labelled ATF2 BD and IMP α paralogs.

Comment 6: Figure 1D: Label the N- and C-termini of Imp α and ATF2, and include labels for the ARM repeats.

We thank the reviewer for this helpful suggestion. Figure 1D has now been revised to label the N- and C-termini of both IMP α and ATF2, and to indicate the ARM repeat architecture of IMP α for improved structural clarity.

Comment 7: Figures 3C, 4C, 5B, and 6B: Consider replacing pie charts with bar charts to improve visualization of ratio data.

We appreciate the reviewer's suggestion. We have previously presented similar IF quantitative data using pie charts in several of our published studies, where this format was well received by readers and found to be clear and intuitive for representing proportional data. Based on this experience, we believe that pie charts remain an informative and effective way to visualise the ratios shown in Figures 3C, 4C, 5B, and 6B. Therefore, we have retained this format in the current manuscript.

Comment 8: Include a supplementary figure showing the electron density for the ATF2 peptides in the crystal structures.

The image has been added to supplementary as requested by reviewer (see Supplementary Figure 2).

Editors comments:

-Please provide scale bar information in the legend for Figure 3B and Figure 4B.

-We have added the requested information

-Please provide details for objective used (including objective type and numerical aperture) under the description of imaging method.

We have included the requested information in the Materials and Methods.

-Please clarify in the methods if imaging of GFP fusion proteins was done with the help of any antibodies and if so, provide details of antibodies used.

No antibody was used.

-Please add a Category and Keywords for your manuscript in our system.

-Please add the X and Bluesky handles of your host institute/organization, as well as your own and/or one of the authors, in our system.

January 26, 2026

RE: Life Science Alliance Manuscript #LSA-2025-03543-TR

Dr. Jade K Forwood
Charles Sturt University
School of Biomedical Sciences
Boorooma St
Wagga Wagga 2650
Australia

Dear Dr. Forwood,

Thank you for submitting your Research Article entitled "A non-canonical in trans mechanism and paralog-selective recognition governs ATF2 nuclear import.". It is a pleasure to let you know that your manuscript is now accepted for publication in Life Science Alliance. Congratulations on this interesting work.

Your manuscript will now progress through copyediting and proofing. At the proofing stage, we encourage you to completely address the concern of Reviewer 2 on clearly indicating all the fluorescently labelled molecules for Figures 1C and 2.

It is journal policy that authors provide original data upon request.

DISTRIBUTION OF MATERIALS:

Again, congratulations on a very nice paper. I hope you found the review process to be constructive and are pleased with how the manuscript was handled editorially. We look forward to future exciting submissions from your lab.

Sincerely,

Sarita Hebbar, PhD
Scientific Editor
Life Science Alliance
<http://www.lsajournal.org>